# A Narrative Review on REM Sleep Deprivation: A Promising Non-Pharmaceutical Alternative for Treating Endogenous Depression

**DOI:** 10.3390/jpm13020306

**Published:** 2023-02-10

**Authors:** Cătălina Angela Crișan, Zaki Milhem, Roland Stretea, Ioan-Marian Țața, Răzvan Mircea Cherecheș, Ioana Valentina Micluția

**Affiliations:** 1Department of Neurosciences, Psychiatry and Pediatric Psychiatry, Faculty of Medicine, Iuliu Hațieganu University of Medicine and Pharmacy, 400012 Cluj-Napoca, Romania; 2Clinical Hospital of Infectious Diseases, 400348 Cluj-Napoca, Romania; 3Automatics and Computers Doctoral School, Politehnica University of Bucharest, 060042 Bucharest, Romania; 4Department of Public Health, College of Political, Administrative and Communication Sciences, Babeș-Bolyai University, 400294 Cluj-Napoca, Romania

**Keywords:** endogenous depression, REM sleep deprivation, personalized medicine

## Abstract

Endogenous depression represents a severe mental health condition projected to become one of the worldwide leading causes of years lived with disability. The currently available clinical and non-clinical interventions designed to alleviate endogenous depression-associated symptoms encounter a series of inconveniences, from the lack of intervention effectiveness and medication adherence to unpleasant side effects. In addition, depressive individuals tend to be more frequent users of primary care units, which markedly affects the overall treatment costs. In parallel with the growing incidence of endogenous depression, researchers in sleep science have discovered multiple links between rapid eye movement (REM) sleep patterns and endogenous depression. Recent findings suggest that prolonged periods of REM sleep are associated with different psychiatric disorders, including endogenous depression. In addition, a growing body of experimental work confidently describes REM sleep deprivation (REM-D) as the underlying mechanism of most pharmaceutical antidepressants, proving its utility as either an independent or adjuvant approach to alleviating the symptoms of endogenous depression. In this regard, REM-D is currently being explored for its potential value as a sleep intervention-based method for improving the clinical management of endogenous depression. Therefore, this narrative review represents a comprehensive inventory of the currently available evidence supporting the potential use of REM-D as a reliable, non-pharmaceutical approach for treating endogenous depression, or as an adjuvant practice that could improve the effectiveness of currently used medication.

## 1. Introduction

Sleep is a reversible physiological state characterized by a complex pattern of cerebral electrical activity. Once the wakefulness state is suppressed, the normal sleep cycle that follows is composed of two distinct, yet alternating, phases called non-rapid eye movement (NREM) and REM [1,2,3]. Normal REM sleep is mainly associated with dreaming, characterized by fast eye movements, mixed-frequency electroencephalographic rhythm, and muscle atonia. This common paralysis of the skeletal muscles has a protective role, as it obstructs the development of complex physical movements during REM sleep [4]. 

Several research studies have noticed a strong interplay between cholinergic and monoaminergic neurons in the brainstem, which form a complex intercellular relationship that appears to regulate the activation of REM sleep [5,6,7]. Among the most important neurotransmitters involved in the generation and maintenance of sleep are the biogenic amines (norepinephrine and serotonin). Although these essential neurotransmitters participate in the initiation of each sleep phase, both are at their lowest during REM sleep [8]. Disturbances of norepinephrine and serotonin systems may contribute to REM sleep abnormalities in different conditions, including endogenous depression [9] and anxiety [10]. 

Recently, a growing body of studies has emerged emphasizing the association between REM sleep behavior and different endogenous depression-associated symptoms, thus highlighting the diagnostic value of dysregulated REM sleep patterns [10,11,12,13]. Most depressed patients suffer from sleep abnormalities. Wang et al. reported that endogenous depression-induced sleep irregularities included a decrease in REM sleep latency, however, there was also an increase in REM sleep duration and density. Hence, in the sleep science community, REM sleep alterations started to be considered essential biomarkers for predicting the risk of endogenous depression. The researchers have also found a consistent clinical association between altered norepinephrine–serotonin systems and REM sleep abnormalities in patients with endogenous depression [14]. Likewise, such findings were confirmed by other similar studies [15,16]. Hence, REM sleep pattern disruption is considered to be related to several psychiatric disorders, including endogenous depression and anxiety [17,18,19], which also confirms its potential as a diagnostic biomarker. REM-D can be defined as a repertoire of pharmaceutical and non-pharmaceutical approaches designed to reduce overall REM sleep duration. Although there is an association between total sleep deprivation and the impairment of several emotion- and cognition-based functions, including decision-making [20], perceived emotional intelligence, constructive thinking skills [19], moral judgement [18], and reactivity toward negative stimuli [20], there is currently no evidence linking these side effects with REM-D. In addition, almost all antidepressants influence sleep patterns, mainly by suppressing REM sleep. Hence, REM-D is considered the underlying mechanism of most pharmaceutical-based antidepressants and a valuable indicator of their efficacy [21,22,23,24].

The development of endogenous depression was recently described as a combination of two key factors: reduced levels of cerebral monoamines (particularly norepinephrine and serotonin) and prolonged periods of REM sleep. Thus, REM-D started to be explored as a non-drug treatment for endogenous depression [25,26].

This narrative review summarizes the current experimental and clinical evidence supporting the effectiveness of REM-D as a non-pharmaceutical approach for alleviating endogenous depression-associated symptoms, with a side focus concerning REM-D protocol optimization and associated risks. 

## 2. Methods

We conducted a comprehensive search in PubMed (accessed on 10 January 2023 https://pubmed.ncbi.nlm.nih.gov/) to identify experimental studies presenting empirical findings that support the utility of REM-D as an alternative approach for alleviating endogenous depression-associated symptoms. We searched for academic scientific papers, starting with the pioneering work of Vogel et al. from 1972 up to the present time, whose abstracts included a combination of the following terms: “depression; or endogenous depression” and “rapid eye movement; or REM”, with or without the term “deprivation”. The papers had to meet the following inclusion criteria: (i) the paper reported empirical results on the beneficial use of REM-D in endogenous depression; and (ii) the paper reported empirical results on the detrimental use of REM-D in endogenous depression or reported the observed side effects, if any. In terms of exclusion criteria, as this narrative review focuses on all the experimental and clinical findings, regardless of the study model, concerning the use of REM-D as an innovative and non-invasive approach for alleviating depression-associated symptoms—either alone or as an adjuvant—we excluded scientific articles that lacked empirical findings.

The scientific articles that passed the inclusion criteria allowed us to structure this narrative scientific review article into 4 comprehensive sections (Current view on endogenous depression; Physiology and pathology of REM sleep; REM sleep deprivation as a non-pharmaceutical choice for treating endogenous depression; Risks and side effects associated with REM sleep deprivation) and to contextualize our theories regarding the benefits of REM-D as a good method for improving depression-associated symptoms. 

The scientific papers that were included in this narrative review were closely examined by three reviewers (C.A.C., I.V.M., and Z.M.). Each selected paper was assessed and the following variables were examined: original or review data, study design, number of patients or study models, confirmed disease, medication (if any administered), study period, and beneficial or detrimental effects on REM-D. Any disagreements during the writing of the article were settled reaching consensus between two reviewers (C.A.C., I.V.M., and Z.M.).

## 3. Current View on Endogenous Depression

Endogenous depression represents a common, yet serious psychiatric condition. Characteristic manifestations of endogenous depression include loss of interest in activities generally considered pleasurable, desolation, irritability, feelings of worthlessness, hopelessness, guilt, concerns over death, suicidal ideation, and sleep disturbances [27,28,29]. These symptoms affect how depressive patients feel, think, and handle daily activities, leading to impaired social connections, reduced work productivity, and a massive decrease in life quality [30,31,32]. 

Based on the circumstances underlying its development, there are different types of depression: major depression (unipolar depression), the most common type characterized by at least two weeks of symptoms that typically interfere with one’s ability to sleep, eat, and work; persistent depressive disorder (dysthymia) with less severe symptoms but which usually last for at least two years; perinatal depression, which occurs when a woman experiences significant depression-associated symptoms during pregnancy or postpartum; seasonal affective disorder, when the patient feels depressed in relation to a particular season, usually late fall and early winter; and depression with psychosis symptoms, characterized by severe manifestations, such as delusions or hallucinations [33,34,35].

Although underdiagnosed, endogenous depression is a severe condition that could pave the ground for other complementary diseases, especially since depressive disorders are ranked as the third leading cause of years lived with disability worldwide [36,37]. In fact, Steffen et al. observed that almost all mental disorders are at least twice as prevalent in individuals suffering from endogenous depression, with a severity-dependent response relationship. The most pervasive somatic depression-related comorbidities are dorsopathies, hypertensive diseases, and metabolic disorders. In addition, a two to threefold higher prevalence of neurological diseases, including sleep disorders, migraine, and epilepsy, was also noticed [38]. Nonetheless, endogenous depression is strongly correlated with suicidal ideation and attempt, exhibiting a high suicide risk rate of approximately 15%. This is, by far, the most adverse outcome, as over 700,000 people die by suicide every year [39,40].

From an epidemiological point of view, the prevalence of depression (in all age groups) has increased in the past few decades [41,42]. According to the latest statistics available from WHO, endogenous depression affects about 3.8% of the world population, so approximately 300 million people are dealing with depressive symptoms, making it a leading cause of disability [43,44,45,46]. The lifetime prevalence of depressive conditions ranges between 10 and 20% in US adults [47]. However, the number of depressive patients may be a lot higher due to the arrival of the COVID-19 pandemic, which caused a 25% increase in the already growing number of depressive individuals [48,49,50]. In addition, the increased incidence of somatic manifestations that can lead to more frequent utilization of primary care, urgent care, and emergency or inpatient services, was also noted in depressed individuals [51,52,53,54,55]. As an effect, the associated economic burden has substantially increased in the past few decades, reaching over USD 326.2 billion in the USA alone [56]. In total, poor mental health was estimated to cost the world economy roughly USD 2.5 trillion annually, an expense projected to rise to USD 6 trillion by 2030 [57].

In parallel with the constant growth of endogenous depression cases, there was a significant increase in the pharmaceutical-based antidepressant repertoire following the introduction of selective serotonin reuptake inhibitors (SSRIs). However, the currently available drugs present with moderate efficacy relative to placebo, relatively slow onset of action, possible withdrawal symptoms, treatment resistance, and problems with compliance [46,58,59,60,61]. Moreover, in low- and middle-income countries, over 75% of depressed patients have limited or no access to proper treatment due to social stigma, a shortage of medical resources, and a lack of trained psychotherapists. In addition, some depressive patients remain undiagnosed, which markedly contributes to the already growing number of severe cases and the overall treatment costs [62,63,64,65,66]. 

Taken together, the constantly growing incidence of endogenous depression, the inconveniences associated with the currently available therapeutic options, and the related economic burden, create a clear demand for the development of more convenient ways to spot and alleviate the symptoms of endogenous depression [67,68]. 

## 4. Physiology and Pathology of REM Sleep

Regular sleep is usually defined as a reversible state of body and mind disconnection, where the nervous system is relatively inactive, the skeletal muscles are relaxed, the metabolism is reduced, the sensory responses are decreased, and the consciousness is practically suspended [69,70]. In the past few decades, advances in electroencephalography (EEG) [71] and polysomnography (PSG), considered the gold standard for diagnosing sleep disorders, have allowed a more in-depth characterization of sleep architecture [72]. Thus, we now know there are two alternating phases of sleep: NREM and REM sleep [73].

A sleep episode starts with a period of NREM sleep, progressing through all of its four stages, followed by a phase of REM sleep. However, we do not remain in REM sleep until wakefulness; rather, we cycle between the stages throughout the night. NREM sleep accounts for about 75–80% of total sleep time, while REM sleep constitutes the remaining 20–25%. The average length of the first NREM–REM sleep cycle is 70–100 min, but in healthy individuals, this cycle increases during the night [74].

REM sleep is generally associated with various pathological and psychological phenomena [75]. The research on REM sleep dates back to 1953, when Kleitman and Aserinsky studied human infants and observed that periods of profound sleep were associated with rapid eye movements and alternated with quiescent sleep periods [76]. A few years later, in 1957, Kleitman teamed up with Dement and saw that REM sleep periods were associated with specific brain-wave patterns and dreaming [77,78]. 

In addition to the REM produced by the bursting of oculomotor muscles in healthy humans, REM sleep is also characterized by a reduced amplitude and greater frequency of cortical EEG waves. This is suggestive of waking, high-amplitude theta waves in the hippocampal EEG; active suppression of skeletal muscle activity; intermittent muscle twitches; autonomic and respiratory activation; fluctuations in brain/body temperature; and an elevated arousal threshold [79].

REM sleep genesis was first investigated by Jouvet and Michel in 1960, who identified a brain region in the dorsal pontine brainstem involved in generating muscle atonia during REM sleep [80]. Subsequent research defined this region as the sublaterodorsal nucleus (SLD) [81,82,83]. Currently, we know that the ventral portion of the SLD contains a substantial population of spinally projecting neurons that function to produce the motor atonia of REM sleep. The SLD neurons, which are hyperactive in REM sleep, are glutamatergic and produce motor atonia by activating a set of inhibitory interneurons in both the ventral medulla and the spinal cord [82,84,85]. In addition, besides the regulation of REM sleep atonia, glutaminergic cLDT-SLD neurons are also involved in the generation of the forebrain features of REM sleep. As such, the selective and acute activation of cLDT-SLD glutamatergic neurons can potently drive REM sleep [86,87]. CLDT-SLC contains two sets of glutamatergic neurons that confer a dual, segregated functionality on CLDT-SLC. As such, the reticulospinal REM sleep atonia-generating neurons promote corticohippocampal activation during REM sleep, while the parabrachial nucleus—the medially adjacent precoeruleus region—regulates REM sleep duration [81,82,88,89]. Even if the regulatory mechanism of SLD neurons is not fully deciphered, many different sources of SLD-directed synaptic inputs have been identified, including acetylcholine, noradrenaline, serotonin, GABA, and glutamate [90,91,92,93,94].

Throughout the years, REM sleep has been linked with important neurodevelopment and neuromodulator functionalities. Thus, it is believed that REM sleep stimulates brain development, particularly motor learning, and sensory system development [95,96,97]. 

One of the most supported REM sleep functionalities is enabling memory formation and consolidation [98,99]. For example, Li et al. showed that REM sleep appears to selectively prune and maintain new synapses associated with particular types of motor learning while also facilitating learning and memory consolidation [100]. In this case, several REM sleep functions were described, including sustaining cortical plasticity [101,102], restoring aminergic receptor function [103], and heightening general creativity [104,105].

REM sleep behavior disorder (RBD) is a neurological condition that causes atonia, resulting in excessive motor behaviors during REM sleep [82,106,107]. The worst aspect of RBD is that most patients develop a neurodegenerative disease within 6–15 years of initial RBD diagnosis [108]. Narcolepsy is another common sleep disorder associated with deviations of REM sleep behavior, caused by the loss of hypothalamic orexin cells and characterized by excessive sleepiness, disturbed REM sleep, sleep paralysis, and hypnagogic hallucinations [109]. Another REM sleep-associated disorder is cataplexy, which is an emotion-driven condition, highlighting the link between emotional processing and muscle atonia during REM sleep [110,111]. 

## 5. REM Sleep Deprivation as a Non-Pharmaceutical Choice for Treating Endogenous Depression

Some of the first investigations conducted on the effects of REM-D in the treatment of endogenous depression date back to the early 1970s and start with the pioneering work of Vogel et al. They designed a research protocol to study the hypothesis that the symptoms of endogenous depression could be relieved by increased REM pressure, defined by the authors as an increase in REM sleep produced by REM-D via awakening. Their work proved that increasing REM pressure by the administration of an external agent (such as monoamine oxidase inhibitors or tricyclic antidepressants) decreases REM sleep and REM-D by awakening at the start of each REM period. The scientists reported that after experiencing increased REM pressure due to REM-D, five out of eight depressed patients improved markedly and one patient improved slightly, while the treatment had no effect on the remaining two subjects. Based on these results, Vogel et al. suggested that REM pressure may be the mechanism behind the effectiveness of most antidepressant drugs [112]. At the beginning of the next decade, Vogel et al. gathered additional evidence by comparing sleep variables in 14 drug-free endogenous depressive subjects and 14 age- and insomnia-matched, non-depressed controls before and after REM-D by awakening, thus strengthening his hypothesis that antidepressant drugs alleviate endogenous depression-associated symptoms by REM-D [113]. Three years later, Vogel formulated a set of criteria that validated REM-D as the primary mechanism of action underlying the effectiveness of antidepressant drugs [114]. 

Rosales-Lagarde and her group of sleep researchers conducted a study designed to assess the effects of REM-D on emotional reactivity to threatening visual stimuli in a cohort of 20 adult, male volunteers between 21 and 35 years of age. Subjects in the REM-D group were kept awake for 2 min every time the PSG showed slow-wave activity. Sleep spindles and K complexes were no longer present in the EEGs, which, instead, were characterized by low-voltage fast activity accompanied by decreased EMG activity. This procedure reduced REM sleep to only 4% of total sleep time. Their findings showed an enhancement of emotional reactivity after REM-D in humans [115], which has been positively correlated with improved symptoms in patients with a depressive disorder [116].

In a separate study conducted by Cartwright et al., the contribution of controlled REM-D upon remission from untreated endogenous depression was investigated over five months in a cohort of 20 depressed subjects compared with 10 control volunteers. Surprisingly, at the end of the study, 60% of the individuals from the depressed group entered remission, admitting improved levels of self-reported symptoms. These findings support the utility of REM-D as an effective tool in the non-drug management of endogenous depression-related symptoms [117]. 

A recent study by Ju et al. investigated the mechanisms underlying the antidepressant effects of REM-D and fluoxetine, a selective serotonin reuptake inhibitor, in a depressive rat model. The researchers reported an enhanced repertoire of benefits, including increased body weight, prompted behavior, and some cellular protective effects, such as alleviating endogenous depression-induced damage, attenuating apoptosis, and maintaining A1 adenosine receptor activity. Hence, these findings indicate an adjuvant role for REM-D, when induced in combination with fluoxetine, for practical use against endogenous depression [118].

Besides its antidepressant efficacy, REM sleep fragmentation was closely associated with depressive status after a study conducted on 54 depressed patients with short-term insomnia disorder. Wu et al. developed a REM sleep fragmentation-based regression model that could predict the risk of endogenous depression with an 83.7% prediction accuracy, thus promoting REM as a viable index for estimating depression risk and a biomarker for treatment response [119].

A comprehensive summarization of these studies is further presented in Table 1.

Over the past few years, the sleep science community has extensively studied the neurological links between electroencephalographic biomarkers, including REM sleep behavior and psychiatric disorders, particularly endogenous depression [14,123]. As such, Wu et al. conducted a research study on 54 depressive patients with short-term insomnia disorder and assessed their REM sleep latency, REM sleep arousal index, and NREM sleep arousal index. After three months of follow-up, it was noted that the total Beck endogenous depression inventory (BDI) was positively correlated with REM sleep fragmentation and negatively correlated with REM sleep latency. Then, using linear regression, they generated a regression model that could predict the risk of endogenous depression with 83.7% accuracy. These findings, together with other pioneering work, support the use of REM sleep behavior as a viable endogenous depression predictor marker, indicating that REM-D could also predict the therapeutic outcome [119,124]. 

## 6. Risks and Side Effects Associated with REM Sleep Deprivation

To ensure the safety of such non-pharmaceutical practice, several studies evaluated the possible side effects associated with REM-D. In this regard, Casey et al. reported that over two nights of induced REM-D, there were no adverse effects on short-term or working memory, neither on verbal implicit memory nor on the overall memory performance [125]. Similar results were obtained by Morgenthaler et al. in a research study on REM-D where there was no difference in the recognition accuracy (neutral and emotional) identified between the study (REM-D-induced) and control group, thus confirming that REM-D did not influence memory consolidation [126]. In addition, Mathangi et al. showed that although 96 h of REM-D might cause an increase in oxidative stress levels, 24 h of restorative sleep will completely reverse this effect [127]. Therefore, there is currently no scientifically proven evidence to clearly associate REM-D with any negative side effects or associated health risks.

Currently, the long-term consequences of REM-D are unknown. However, some of the research conducted in animal models pinpoint a series of detrimental effects associated with prolonged REM-D that extend to a molecular level, such as increased oxidative stress, spatial memory impairment, and behavioral and performance alterations [127,128]. However, extensive investigations are required in order to confirm these side effects in humans and establish a clear relationship with REM-D interventional method, frequency, and duration. 

## 7. Conclusions

This narrative review article provides a comprehensive overview of the latest evidence supporting the potential use of REM-D as a putative, non-pharmaceutical antidepressant or adjuvant practice that could improve the effectiveness of currently used medication. Endogenous depression is currently a major health problem with severe, often fatal, outcomes. This condition represents a heavy burden for both the patient and the caregiver, so, as it becomes more prevalent year by year, the need for effective and convenient treatment approaches is urgent [129,130]. Hence, this paper provides a status check regarding endogenous depression epidemiology and sleep science, with a central focus on REM sleep. Taken together, all the presented evidence, particularly the growing incidence of endogenous depression, the lack of convenient, yet effective therapeutic strategies, and the proven potential of REM-D as an antidepressant, corroborates a significant body of evidence that supports the further use of REM-D in the development of innovative solutions that could help spot and alleviate endogenous depression-associated symptoms. However, the future direction regarding the translation of REM-D approaches in the clinical management of endogenous depression should procced only after extensive validation on larger cohorts of human patients and a comprehensive assessment of the long-term side effects of REM-D.

## Figures and Tables

**Table 1 jpm-13-00306-t001:** Studies supporting the efficacy of REM-D as a non-drug antidepressant (ED—endogenous depression; RD—reactive depression).

Study	Study Model	REM-D Method	Duration	Conclusions	Refs.
Vogel et al., 1972	12 EDs (seven experimental, five controls) 12 EDs (eight experimental, four controls)	Recurrent awakening during REM sleep	Up to 13.6 weeks	REM-D relieves the symptoms of ED REM pressure is the mechanism behind most antidepressant drugs	[112]
Vogel et al., 1980	14 drug-free EDs 14 matched controls	Recurrent awakening during REM sleep	Up to 13.6 weeks	REM-D improved depression to the extent that it stimulated the oscillator and corrected one manifestation of circadian rhythm disruption	[113]
Vogel, 1983	34 EDs (17 experimental, 17 controls) [120] 18 RDs (11 experimental, 7 controls) [120] Data from Imipramine-treated patients from the British Medical Research Council 1965 [121]	Recurrent awakening during REM sleep Imipramine-treated patients from the British Medical Research Council 1965 [121]	24 weeks	REM-D is the mechanism of action of antidepressant drugs	[114]
Rosales-Lagarde et al., 2012	20 right-handed adult male volunteers between 21–35 years of age (12 REM-D and 8 NREM-I)	Recurrent awakening during REM sleep	Four nights (one night for treatment)	Post-REM-D emotional reactivity, which has been positively correlated with improved ED symptoms	[115,116]
Cartwright et al., 2003	20 depressed subjects compared with 10 control volunteers	Recurrent awakening during REM sleep	Five months	60% of the ED group entered remission. Hence, REM-D could be a non-drug antidepressant	[117]
Ju et al., 2021	Depressive male Sprague–Dawley rat model	Recurrent awakening during REM sleep, which reduced REM sleep to only 4% of total sleep time	28 days	These findings indicate an adjuvant role of REM-D when in combination with the administration of fluoxetine	[118]
Wu et al., 2021	54 depressed patients with short-term insomnia	REM sleep fragmentation	Three months	REM sleep is a characteristic marker for assessing the risk of ED	[119]
Maudhuit et al., 1996	Depressive male Sprague–Dawley rat model	Zimelidine dissolved in 1 mL saline was injected twice a day at a dose of 2.5 mg/kg IP for 14 days. On day 15, only the morning dose was administered. Control rats received 1 mL saline REM-D by placing the rats on a platform fenced by water Control rats stood on a platform where they could lie down for REM sleep	Zimelidine twice a day for 14 days, once on the 15th day. Four successive REM-D sessions	Electrophysiological activity of 5-HT neurons in the nucleus raphe dorsalis revealed that chronic treatment with both zimelidine and REM-D induced hyporeactivity of 5-HT neurons to the inhibitory effect of depression-like citalopram administration	[122]

## Data Availability

Not applicable.

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
