# Peer review of "A Narrative Review on REM Sleep Deprivation: A Promising Non-Pharmaceutical Alternative for Treating Endogenous Depression"

_jpm, 2023, doi:10.3390/jpm13020306_

Round 1
Reviewer 1 Report (Previous Reviewer 3)
The Review of the Article
“A Narrative Review on REM Sleep Deprivation: A promising Non-Pharmaceutical Alternative for Treating Endogenous Depression”
(Manuscript jpm-2187758)
The current descriptive review is both a check of the state of endogenous depression and REM sleep, and, most importantly, a comprehensive list of evidence supporting the potential use of REM sleep deprivation (REM-D) as a proposed, non-invasive and non-invasive method. A growing number of experimental studies confidently support REM sleep deprivation as a real mechanism underlying the action of most drug-based antidepressants, proving its usefulness as an independent or auxiliary approach to alleviate the symptoms of endogenous depression. Thus, the current descriptive review is both a check of the state of endogenous depression and REM sleep, and, most importantly, a comprehensive list of evidence supporting the potential use of REM sleep deprivation (REM-D) as a presumptive and non-invasive method.
During the analysis of the materials of the article “A Narrative Review on REM Sleep Deprivation: A promising Non-Pharmaceutical Alternative for Treating Endogenous Depression”, I come to the conclusion that the article title and abstract are appropriate.
The purpose of the article and its significance is stated clearly. The study methods are sound and appropriate. The writing is clear and concise. The conclusions are accurate and supported by the content. The article is of interest to members of the education research community. The study was conducted according to the guidelines of the Declaration of Helsinki, and approved by the Regional Ethics Review Board.
I recommend the article “A Narrative Review on REM Sleep Deprivation: A promising Non-Pharmaceutical Alternative for Treating Endogenous Depression” for publication.
Professor, Head Specialist (Research Fellow),
V. Serbsky National Medical Research Centre of Psychiatry and Addiction, Russian Federation;
Professor, Institute of Psychology,
Belarusian State Pedagogical University (Minsk)
S.Igumnov /Sergey Igumnov/
Author Response
Dear reviewer, thank you for taking the time to revise our work and your valuable feedback!
Sincerely,
Zaki Milhem MD, PhD(c)
Reviewer 2 Report (New Reviewer)
The manuscript titled “A Narrative Review on REM Sleep Deprivation: A Promising Non-Pharmaceutical Alternative for Treating Endogenous Depression” aimed to review experimental studies that support the utility of REM-D as an approach for alleviating endogenous depression. This comprehensive review indicates the potential use of REM sleep deprivation (REM-D) as a putative, non-invasive, and non-pharmaceutical approach for diagnosing and treating endogenous depression. However, there are several concerns in the manuscript.
Comments
#1. The descriptions of first half of the abstract are general explanations about endogenous depression. Abstract is expected to be more focused on the relationship between REM-D and endogenous depression.
#2. Chapters 3 and 4 look verbose and should be shortened.
#3. The terms "REM sleep deprivation" and "REM pressure" can be difficult to understand for the first-time readers. Therefore, the authors should provide more detailed descriptions about these terms.
#4. References 121-123 don't seem to appear in the text.
#5. The authors should revise Table 1 to be more self-explanatory. In addition, I recommend that the names of first authors and the year of publication should be added to Table 1.
#6. In the chapter 6, the authors described the risks and side-effects associated with REM-D. Are there any previous studies that mentioned long-term risks associated with REM-D?
#7. The authors may mention the future directions of the REM-D as an approach for endogenous depression.
Author Response
Response to Reviewer
#1. The descriptions of first half of the abstract are general explanations about endogenous depression. Abstract is expected to be more focused on the relationship between REM-D and endogenous depression.
Dear reviewer, thank you for your valuable feedback! We revised the entire abstract and refined it to be more focused on the relationship between REM-D and endogenous depression.
#2. Chapters 3 and 4 look verbose and should be shortened.
Dear reviewer, thank you for your valuable feedback! We revised both chapter 3 and 4 and made them less verbose and, hopefully, much clearer now.
#3. The terms "REM sleep deprivation" and "REM pressure" can be difficult to understand for the first-time readers. Therefore, the authors should provide more detailed descriptions about these terms.
Dear reviewer, thank you for your valuable feedback! We provided clear explanations for each term, so it is easier to understand for the first-time readers now.
#4. References 121-123 don't seem to appear in the text.
Dear reviewer, thank you for your valuable feedback! Actually, they do appear in the text. Please check the Table 1, column 2 and 3, row 5, respectively column 6, row 10.
#5. The authors should revise Table 1 to be more self-explanatory. In addition, I recommend that the names of first authors and the year of publication should be added to Table 1.
Dear reviewer, thank you for your valuable feedback! We revised Table 1 and made it more self-explanatory. We also added the names of the first authors and the year of publication as a first column in our table.
#6. In the chapter 6, the authors described the risks and side-effects associated with REM-D. Are there any previous studies that mentioned long-term risks associated with REM-D?
Dear reviewer, thank you for your valuable feedback! We added an additional paragraph regarding the possible long-term side-effects of REM-D based on the available studies conducted in animal models.
#7. The authors may mention the future directions of the REM-D as an approach for endogenous depression.
Dear reviewer, thank you for your valuable feedback! We included the future directions of the REM-D as an approach for the management of endogenous depression in our Conclusion chapter.
Thank you for taking the time to revise our work!
Round 2
Reviewer 2 Report (New Reviewer)
Thank you for addressing all the comments. The manuscript has been generally improved; however, some typos seem to have remained. My recommendation is to accept it after the elaborate spellcheck.
This manuscript is a resubmission of an earlier submission. The following is a list of the peer review reports and author responses from that submission.
Round 1
Reviewer 1 Report
Thank you for the opportunity to review this study entitled “REM Sleep Deprivation: A Promising Non-Pharmaceutical Alternative for Treating Depression.” (jpm-2069792).
The present paper presents a review concerning depression and REM sleep, highlighting the potentiality of sleep deprivation as alternative non-pharmaceutical treatment.
In my opinion, although the research topic is relevant and interesting, the paper does not meet the standards for publication in a peer-reviewed journal.
For example, detailed information is not provided on the method, the type of articles selected (and why), the reliability of the sources, inclusion-exclusion criteria, and so on.
These issues lead me to believe that the paper is not suitable for publication.
Therefore, with regret, I suggest rejecting the article.
Best wishes
Author Response
Response to Reviewer 1 Comments
Point 1: The present paper presents a review concerning depression and REM sleep, highlighting the potentiality of sleep deprivation as alternative non-pharmaceutical treatment.
In my opinion, although the research topic is relevant and interesting, the paper does not meet the standards for publication in a peer-reviewed journal.
For example, detailed information is not provided on the method, the type of articles selected (and why), the reliability of the sources, inclusion-exclusion criteria, and so on.
These issues lead me to believe that the paper is not suitable for publication.
Therefore, with regret, I suggest rejecting the article.
Response 1: Dear reviewer,
Thank you for your valuable feedback.
Although we respect your opinion, please note that our paper is, in fact, a narrative review, and not a systematic one, nor a metanalysis. Hence, our work is meant to corroborate some of the most consisting and reliable scientific findings supporting the chosen topic, which is the potential value of REM sleep deprivation as a non-pharmaceutical alternative for alleviating endogenous depression-associated symptoms. Therefore, there was no need to further detail our methodology, source reliability, nor their inclusion-exclusion criteria.
We would appreciate our work to be considered and evaluated as what it is, a narrative review.
Thank you for taking the time to revise our work!
Reviewer 2 Report
Overall the authors present interesting work on the use of REM sleep deprivation as a complementary practice to improve depressive symptoms. However, the following are some issues that the authors should consider.
Title: Since a narrative review has been performed, this should be indicated in the title. Likewise, throughout the text, when mention is made of the type of methodology used (as in line 279), this should be indicated again.
Abstract: The authors make a bold statement when they say that " The currently available therapies designed to alleviate the symptoms associated with depression have several limitations, especially the lack of effectiveness." For example, there is evidence of significant reduction of depressive symptoms through Mobile Phone-Based interventions (Astafeva et al., 2022), some psychological therapies (MBCT, LTPP, and IPT in combination with medication) have a moderate positive effect on patients with persistent forms of depression (McPherson & Senra, 2022) even acupressure appears to be effective in reducing depression in people with mild-to-moderate primary and secondary depression (Lin et al., 2022). I personally find it too bold to make such a clear-cut statement in the title, in a context as changing and constantly updated as that of depression. I recommend the authors to rephrase this part.
In any case, if in the development of the article they refer to "endogenous depression" as can be read in the introduction, this should also be reflected in the abstract.
Lines 51-53: The phrase " Finally, the authors confirmed the clinical link between dysregulated norepinephrine and serotonin systems and REM sleep abnormalities in depression" is misleading as constructed. I recommend rephrasing.
Line 61: Clarify what is meant by "REM-D" (assumed to refer to REM sleep deprivation but should be stated beforehand)
Lines 125-133: The authors contradict what they state in the article and the references they use for it, as well as contradicting what is stated in the abstract. Some of the references they use to justify their claims are more than 10 years old and there is later evidence that should be used instead, as demonstrated above. In these lines, the authors no longer speak of the lack of effectiveness of the therapies, they state that the therapeutic options are limited and, in addition, they emphasize the economic difficulty as a limitation to access. Without underestimating the central focus of the work, which I find interesting, I would recommend that the authors clarify where exactly they see the problem in relation to currently available therapies for depression, and properly justify it.
Author Response
Response to Reviewer 2 Comments
Point 1: Overall, the authors present interesting work on the use of REM sleep deprivation as a complementary practice to improve depressive symptoms. However, the following are some issues that the authors should consider.
Title: Since a narrative review has been performed, this should be indicated in the title. Likewise, throughout the text, when mention is made of the type of methodology used (as in line 279), this should be indicated again.
Response 1: Dear reviewer,
Thank you for your valuable feedback! We indicated that our work is, in fact, a narrative review everywhere throughout our paper.
Point 2: Abstract: The authors make a bold statement when they say that " The currently available therapies designed to alleviate the symptoms associated with depression have several limitations, especially the lack of effectiveness." For example, there is evidence of significant reduction of depressive symptoms through Mobile Phone-Based interventions (Astafeva et al., 2022), some psychological therapies (MBCT, LTPP, and IPT in combination with medication) have a moderate positive effect on patients with persistent forms of depression (McPherson & Senra, 2022) even acupressure appears to be effective in reducing depression in people with mild-to-moderate primary and secondary depression (Lin et al., 2022). I personally find it too bold to make such a clear-cut statement in the title, in a context as changing and constantly updated as that of depression. I recommend the authors to rephrase this part.
Response 2: Dear reviewer,
Thank you for your valuable feedback! We rephrased the specified sections in our abstract according to your indications.
Point 3: In any case, if in the development of the article they refer to "endogenous depression" as can be read in the introduction, this should also be reflected in the abstract.
Response 3: Dear reviewer,
Thank you for your valuable feedback! We made the revisions and mentioned that we refer to “endogenous depression” everywhere was needed throughout the entire manuscript.
Point 4: Lines 51-53: The phrase " Finally, the authors confirmed the clinical link between dysregulated norepinephrine and serotonin systems and REM sleep abnormalities in depression" is misleading as constructed. I recommend rephrasing.
Response 4: Dear reviewer,
Thank you for your valuable feedback! We rephrase the indicated section in a more concise and clear way.
Point 5: Line 61: Clarify what is meant by "REM-D" (assumed to refer to REM sleep deprivation but should be stated beforehand)
Response 5: Dear reviewer,
Thank you for your valuable feedback! Please note that “REM sleep deprivation” was abbreviated as REM-D even from the abstract.
Point 6: Lines 125-133: The authors contradict what they state in the article and the references they use for it, as well as contradicting what is stated in the abstract. Some of the references they use to justify their claims are more than 10 years old and there is later evidence that should be used instead, as demonstrated above. In these lines, the authors no longer speak of the lack of effectiveness of the therapies, they state that the therapeutic options are limited and, in addition, they emphasize the economic difficulty as a limitation to access. Without underestimating the central focus of the work, which I find interesting, I would recommend that the authors clarify where exactly they see the problem in relation to currently available therapies for depression, and properly justify it.
Response 6: Dear reviewer,
Thank you for your valuable feedback! We made all the clarification required in order set clear the problems associated with the current treatments for endogenous depression, in a manner that, we hope, properly justifies the value of REM-D as a complementary, non-pharmaceutical practice to alleviate depression-associated symptoms.
Thank you for taking the time to revise our work!
Reviewer 3 Report
с
“A narrative review on REM Sleep Deprivation: A Promising Non-Pharmaceutical Alternative for Treating Endogenous Depression”
During the analysis of the materials of this study, presented in the article “A narrative review on REM Sleep Deprivation: A Promising Non-Pharmaceutical Alternative for Treating Endogenous Depression”, I come to the conclusion that the article title and abstract are appropriate.
The purpose of the article and its significance is stated clearly. The study methods are sound and appropriate. The writing is clear and concise. The conclusions are accurate and supported by the content. The article is of interest to members of the clinical and neurophysiological research community. The study was conducted according to the guidelines of the Declaration of Helsinki, and approved by the Regional Ethics Review Board.
I recommend Research Article “A narrative review on REM Sleep Deprivation: A Promising Non-Pharmaceutical Alternative for Treating Endogenous Depression” f or publication on.
Professor, Head Research Fellow,
V. Serbsky National Medical Research Centre of Psychiatry and Addiction, Russian Federation S.Igumnov /Sergey Igumnov/
Round 2
Reviewer 1 Report
Thank you for the opportunity to review the reviewed version of the study entitled “REM Sleep Deprivation: A Promising Non-Pharmaceutical Alternative for Treating Depression.” (jpm-2069792).
I respect the opinion of the authors exposed in their Author response.
However, I strongly disagree and believe that the lack of the possibility of verifying the process, at least in a minimal way, does not allow for evaluating the article suitable for this journal, which reaches higher narrative review standards (e.g., https://doi. org/10.3390/jpm11090849; https://doi.org/10.3390/jpm12060907), in line with other journals (e.g., https://doi.org/10.3389/fpsyt.2021.719490).
Therefore, I consider the paper unsuitable for publication in Journal of personalized medicine.
With the hope that this refereeing does not discourage the authors and stimulates them to produce even better papers.
Best wishes
Reviewer 2 Report
The authors have taken into account the suggestions and comments made in the first round. There are no further comments from my side.